# Coordination games in cancer

**Péter Bayer**[1,2]*, **Robert A. Gatenby**[3], **Patricia H. McDonald**[4], **Derek R. Duckett**[5], **Kateřina Staňková**[6], **Joel S. Brown**[3,7]

1 Toulouse School of Economics, Toulouse, France, 2 Institute for Advanced Study in Toulouse, Toulouse, France, 3 Department of Integrated Mathematical Oncology, Moffitt Cancer Center, Tampa, Florida, United States of America, 4 Department of Cancer Physiology, Moffitt Cancer Center, Tampa, Florida United States of America, 5 Department of Drug Discovery, Moffitt Cancer Center, Tampa, Florida, United States of America, 6 Delft Institute of Applied Mathematics, Delft University, Delft, Netherlands, 7 Department of Biological Sciences, University of Illinois at Chicago, Chicago, Illinois, United States of America

* peter.bayer@tse-fr.eu

## Abstract

We propose a model of cancer initiation and progression where tumor growth is modulated by an evolutionary coordination game. Evolutionary games of cancer are widely used to model frequency-dependent cell interactions with the most studied games being the Prisoner's Dilemma and public goods games. Coordination games, by their more obscure and less evocative nature, are left understudied, despite the fact that, as we argue, they offer great potential in understanding and treating cancer. In this paper we present the conditions under which coordination games between cancer cells evolve, we propose aspects of cancer that can be modeled as results of coordination games, and explore the ways through which coordination games of cancer can be exploited for therapy.

**Data Availability Statement:** All relevant data are within the manuscript and its Supporting information files.

**Funding:** The author(s) received no specific funding for this work.

## 1 Introduction

Cancer cells engage in evolutionary games [1, 2]. For them to do so, they must exhibit two dynamics and certain types of interactions. As an ecological dynamic, cancer cells exhibit survival and proliferation giving rise to changes in their population sizes. As an evolutionary dynamic, the heritable traits of the population change as cancer clades with more successful phenotypes outcompete and replace those with less successful ones [3]. To be an evolutionary game the success of a phenotype must be context dependent. Its success depends upon not just the number but the phenotypes of those cells with which it interacts. Thus, the "best" strategy for a cancer cell will depend upon the strategies of other cancer cells [4]. Researchers have proposed and identified a number of games that may typically occur within a patient's tumor. Cancer cells may engage in the Prisoner's Dilemma [5, 6] and exhibit cooperation [7]. In this case, a cancer cell by co-feeding neighboring cells or by secreting factors that improve the mirco-environment may incur a cost to itself while providing benefits to neighboring cancer cells. If the cooperative cancer cell also derives some benefit from its action then the game shifts from being a Prisoner's Dilemma to a public goods game [8]. By producing VEGF for recruiting vasculature or creating acidic conditions as immune-suppression, the focal cancer cell benefits itself while also benefiting neighbors [9, 10]. Such games can promote a diversity

**Competing interests:** The authors have declared that no competing interests exist.

of cancer cell types, where some become producers while others act as free-loaders, contributing nothing to the public good [11]. Cancer cells may also engage in the Tragedy of the Commons. This happens when the cancer cells over-invest in metabolic pathways, transporters, and capacity so as to pre-empt other cancer cells from acquiring resources from a shared common pool. The interstitial fluids can be the commons, and it may be that cancer cells, for instance, over-express GLUT-1. The level of expression may be higher than would be optimal for the group, but it is advantageous for the individual cancer cell if it gains nutrients that otherwise would have been harvested by its neighbors [12].

Game theory has been suggested as the framework for evolutionarily informed therapies where the physician aims to guide the eco-evolutionary dynamics of cancer towards better outcomes or outright cure [13]. A clinical trial of castrate-resistant metastatic prostate modeled the cancer cells as having three possible strategies in a game that has similarities to a rock-paper-scissors game with three types of prostate cancer cells (T+, T-, TP) promoting or inhibiting each others' fitness in a cyclic fashion [14–16].

### 1.1 Background on coordination games

One class of games that has not been explored in cancer are coordination games. Coordination games describe incentive structures that reward conformity. Deviation by any player results in lower payoffs for the whole population. The most stringent subclass is called *pure coordination* games where positive payoffs are only attainable if all individuals choose the same strategy, for any other strategy combination all individuals receive zero payoffs. A typical application of this game is the adoption a new technology standard. If all participants adopt the new technology, payoffs are maximized, if no one adopts the new technology payoffs are lower but positive, if some participants adopt while others do not, confusion follows and payoffs are zero. If the rewards of coordination are identical, the game is called *choosing sides*. A classic example of this game is driving on either side of the road. If participants all adhere to driving on the same side, the traffic flows and payoffs are positive, otherwise chaos ensues and payoffs are zero. A less stringent class in terms of punishing discoordination is called the *stag hunt* game. One strategy offers low rewards but does not require coordination, the other offers high rewards that can only be attained by coordination. The original version of the game features two hunters who decide on which animal to hunt, stag or hare: if both choose stag, they are able to acquire it through a joint effort for a large reward, if any hunter chooses hare, they are able to acquire it individually for a lower reward, but if one chooses stag while the other chooses hare, the stag hunter fails and receives a payoff of zero.

Evolutionary coordination games replace the element of individual choice with heritable traits and payoffs with reproductive fitness. Populations playing evolutionary coordination games invariably develop similar traits as discoordinating individuals suffer fitness penalties. Furthermore, the populations that achieve coordination more quickly will thrive as a whole, while those that fail to coordinate may fail. Table 1 showcases a coordination game with generic payoff parameters. In SI.1 and SI.2 we outline the mathematical conditions of coordination games.

**Table 1. An evolutionary coordination game with two strategies.** The payoffs satisfy $r_1 \geq r_2 \geq d_1, d_2$. Special cases: Pure coordination ($d_1 = d_2 = 0$), choosing sides ($r_1 = r_2$, $d_1 = d_2 = 0$), stag hunt ($d_1 = 0$, $r_2 = d_2$).

| | | Predominant strategy | |
|---|---|---|---|
| | | Type 1 | Type 2 |
| Focal cell | Type 1 | $r_1$ | $d_1$ |
| | Type 2 | $d_2$ | $r_2$ |

A recent strand of literature classifies 2× 2 evolutionary games of cooperation and defection based on dilemma strength [17–19]. The 'Gamble-Intending Dilemma' (GID) measures the gains a player would realize by defecting rather than cooperating if the opponent cooperates (e.g., in a prisoner's dilemma game, being released from prison). The 'Risk-Averting Dilemma' (RAD) measures the losses a player can avoid by defecting rather than cooperating if the opponent defects (e.g., an extended prison sentence). The two measures can be positive or negative, giving rise to four canonical game classes: If both are positive, defection dominates as in the prisoner's dilemma, if GID is positive but RAD is negative, the game is an anti-coordination game and the dilemma is that of a chicken game (a.k.a. hawk-dove game or snowdrift game), whereas if GID is negative and RAD is positive, it is a coordination game and the dilemma resembles that of a stag hunt game. If both are negative, cooperation dominates, characterizing 'Harmony', that is, the absence of a social dilemma.

Coordination games capture situations without an explicit definition of cooperation or defection. In the purest case, neither strategy is socially optimal, the only thing that matters is whether the players agree. In the general case, the strategies are asymmetric in terms of efficiency (the rewards of players in case of coordination) and risk (the losses of discoordinating players). Similarly, in cancer, neither strategy is a good or bad one; rather its success, depends on the predominant strategy in the population of cancer cells. If type 1 predominates, then a focal cancer cell does best being type 1 and vice versa when type 2 is predominant. This makes identifying coordination games in cancer challenging as the two competing types should not be found together. Their co-occurrence within the same tumor microenvironment or even within the same tumor or patient seems unlikely and would manifest only as a transient dynamics as either type 1 or 2 come to predominate. Furthermore, the predominance of type 1 in one part of a tumor and that of type 2 in another is not strong evidence for a coordination game. For instance, the edge of a tumor may favor 1 and the interior 2 independent of which cell type is initially predominant in that region [20].

In a coordination game it is the predominance of type 1 at the site that makes type 2 unsuitable and vice versa. It is this priority effect that characterizes the coordination game in cancer. Whichever strategy gets established first among the cancer cells excludes the other. This makes it difficult to spot and identify coordination games by observing the outcome within an individual patient. The alternative strategies that the population *could have* coordinated upon are no longer visible. If coordination games exist in cancer, by the time the disease is detected, most cancer patients would likely present with a cancer that has already evolved to a common phenotype. Once clinically detectable, a single strategy from among those possible for the coordination game may have already become fixed within a patient. Thus, the different possible strategies of a coordination game are likely to manifest between rather than within patients. Furthermore, early driver mutations that can vary between patients with the same type of cancer offer a promising starting point for seeking these different strategies. The driver mutation itself along with associated coadapted genes and gene expression may be the principle strategy of a coordination game in cancer.

The paper proceeds as follows: In the remainder of this section we raise examples of possible coordination games in cancer. In Section 2 we build a Lotka-Volterra competition model of cancer where growth is regulated by an underlying coordination game. In Section 3 we investigate the interaction between coordination games of cancer growth and resistance to cytotoxic therapy in a sensitive-resistant model and showcase how therapies that take into account the coordination game can lead to better outcomes. We then showcase how therapy outcomes can be improved further by therapy that increases the transmutation rates between the two competing phenotypes. Section 4 contains the concluding discussion. The Supplementary Information (S1 File) contains technical discussion.

## 1.2 Possible coordination games in cancer

The concept of "driver genes" may help identify a coordination game. In many cancers, a specific gene mutation is observed in virtually all cells and is thought to be a critical event that provides a steady oncogenic signal to maintain survival and proliferation of the cancer cells. Perhaps the best example of this is the oncogenic mutation of Epidermal Growth Factor (EGFR) in lung cancers. This phenotype accounts for 15–20% of clinical lung cancers. Unlike most lung cancer cohorts in which prolonged smoking and exposure to air pollutants are clear risk factors, EGFR-mut cancers tend to occur in younger patients who are non-smokers or have had limited exposure. Typically, EGFR-mut lung cancer cohorts tends to be young, Asian, and female. For these patients, the EGFR-mutation occurs in essentially all of the cancer cells of the primary and metastatic tumors [21]. Furthermore, the overall mutational burden in EGFR-mut lung cancer is significantly smaller than EGFR WT lung cancers, indicating a molecularly more homogeneous intra-tumoral population [22]. We propose that EGFR-mut versus EGFR WT lung cancers represent a coordination game.

Treatment of EGFR-mut lung cancers with tyrosine kinase inhibitors (TKI) that specifically target the EGFR-mut function results in a complete or partial response in about 75% of patients [23]. However, treatment response is transient and tumor progression occurs within 12 to 14 months. By molecular analysis, at least 7 different strategies permit the lung cancer cells to overcome the TKIs (mechanisms in 15% of cases remain unknown). Furthermore, second line treatments with chemotherapy or immunotherapy show minimal efficacy [24]. It has been noted that in many cases, resistance takes the form of the cancer cells losing the EGFR-mut and becoming like EGFR-WT lung cancers. As a coordination game, treating the EGFR-mut may simply shift the cancer to an alternate stable state.

Another subset of lung cancers (about 40%) have an oncogenic KRAS mutations indicating that driver mutations are, to some extent substitutable. But cancer cells with different driver mutations may not be able to coexist within the same patient or tumor. In the case of non-small-cell lung cancer, for example, the driver mutations in KRAS and EGFR seem to be mutually exclusive suggesting that, once a cancer population has one, the other is actively selected against. Thus, one or the other is beneficial, but both are deleterious to the cancer cell. This is necessary for a coordination game but not sufficient. While exhibiting both mutations in a single cell may be selected against, it does not mean that an EGFR mutant cancer cell would be at a disadvantage in a community of KRAS-mutant cancer cells and vice-versa. For these two strategies to be a coordination game, a particular cancer cell with a particular driver mutation would have to be more successful if its neighbors harbored the same driver mutation.

In patients with breast cancer, the ubiquitously expressed beta-arrestin isoforms ($\beta$-arrestin 1; ARRB2, and $\beta$-arrestin 2; ARRB3) may form the biological basis for a coordination game. Beta-arrestins function as "terminators" of G protein-coupled receptor (GPCR) signaling. More recently, $\beta$-arrestins, by virtue of their scaffolding functionality, have also been shown to serve as signal "transducers". As such, $\beta$-arrestins play roles in MAPK signaling [25], and the regulation of several basic cellular functions including cell cycle regulation [26], proliferation, cell migration [27], apoptosis [28], and DNA damage repair [29, 30]. Data supporting a physiologically relevant role for these $\beta$-arrestin-mediated responses are nowhere more compelling than in cancer.

Multiple independent studies have reported a change in the expression of $\beta$-arrestins in breast cancer cells and patient tumors, wherein changes in the expression of $\beta$-arrestins correlate with poor patient survival [31], with $\beta$-arrestin 2 expression serving as a prognostic biomarker in the clinical course of breast cancer. Investigations of several human genomic datasets revealed that the expression of $\beta$-arrestin 1 is downregulated in triple negative breast

cancer, the most aggressive breast cancer subtype in terms of poor outcomes and high rates of relapse [32]. This suggests that cancer cell fitness is maximized by changing the expression of one but not both $\beta$-arrestins. Changing one promotes self-sufficiency in proliferative signaling, while keeping the other $\beta$-arrestin unchanged maintains necessary basic cellular functions.

While these observations can explain how a co-adapted set of genes coordinates cellular processes within a cancer cell, it does not necessarily qualify as a coordination game for explaining why all cells of the patient's cancer show one pattern of changes in $\beta$-arrestin and not the other. The coordination game may result from the way $\beta$-arrestins control aspects of cell-cell signaling through the regulation of GPCR activity, as well as other cell surface receptors. In this case, the value of a particular isoform of $\beta$-arrestin for successful cell-cell communication within the tumor may be determined by the unified expression of this isoform by surrounding cancer cells. That is, the dominant $\beta$-arrestin isoform may define rules of the road for all of the cancer cells.

At this stage, we do not know for certain whether these two examples represent examples of coordination games. They do satisfy necessary conditions in that within a patient there is a predominant type and this predominant type differs between patients that have cancers derived from exactly the same tissue of origin. To be a coordination game, there must be frequency-dependent interactions where the fitness to a cancer cell type depends upon its neighbors' cell types in a manner where being the rare subtype is highly disadvantageous. Once a lung cancer is dominated by EGFR-mut cells has a very different immune (low immune infiltration) and extra-cellular matrix than a KRAS-mut lung cancer with high immune infiltration and less structured extra-cellular matrix. An EGFR-mut cancer cell with its concomitant co-adaptations may find itself greatly disadvantaged within the tumor microenvironment of a KRAS-mut cancer and vice-versa. The same frequency-dependence seems to apply to breast cancer patients where just one or the other $\beta$-arrestin is upregulated but not both.

## 2 Methods

We propose a two-phenotype model of cancer growth with cell types 1 and 2 with the payoffs $r_1, r_2, d_1, d_2$ satisfying the conditions of a coordination game (Table 1).

Our eco-evolutionary setting is described by a pair of ordinary differential equations, modeling the population of the two types. We utilize Lotka-Volterra dynamics to describe our system, developed independently by Alfred Lotka and Vito Volterra in the 1920s. Since then, many extensions of this model have been developed and its equivalence with replicator dynamics with an additional player was proven [33]. Mathematical properties of the classical Lotka-Volterra system show either cyclic oscillation or divergent extinction of one species [34]. In any closed Lotka-Volterra system, the predators will eventually die out with the extinction of the prey. This means that the persistent predator-prey systems, without additional stabilizing mechanisms, should exhibit cyclic oscillations. Here we analyze how in cancer, the additional assumption that cancer cells play coordination games impact their eco-evolutionary dynamics and treatment options. As in most models of cancer growth, the focus of our simulations will be on short-run growth rather than long-run dynamics.

A key difference between our model and Lotka-Volterra competition is that our model's tumor growth depends endogenously on its composition. The model features a joint carrying capacity ($K$) and joint strong Allee-effect (with Allee-threshold $T$) for the cell types. We add type-specific death rates ($c_1, c_2$) to capture the immune system's function of destroying cancer cells and transmutation rates ($m_1, m_2$) capturing cancer's mutation one type to the other. Finally, we account for possible non-linearities of fitness with respect to cell-type frequencies (with convexity parameter $\alpha \geq 1$) in order to capture a more general form of frequency-

**Table 2. The ingredients of our model.**

|  | **Variables** |  | **Parameters** |
|---|---|---|---|
| $x_1(t)$ | Type 1 population | $K$ | Carrying capacity |
| $x_2(t)$ | Type 2 population | $T$ | Allee-threshold |
| $x(t)$ | Tumor size ($x_1(t) + x_2(t)$) | $c_1, c_2$ | Cell death rates |
| $g(t)$ | Tumor composition ($x_1(t)/x(t)$) | $m_1, m_2$ | Mutation rates |
|  |  | $\alpha$ | Convexity parameter |

dependent growth. We offer a detailed discussion on the mechanics and interpretation of the convexity parameter in SI.3. The ingredients of the model are summarized in Table 2.

Our base model is described by the following equations.

$$\dot{x}_1 = x_1 \left( r_1 g^\alpha + d_1 (1-g)^\alpha \right) \left( \frac{x}{T} - 1 \right) \left( 1 - \frac{x}{K} \right) - c_1 x_1 - m_1 x_1 + m_2 x_2, \qquad (1)$$

$$\dot{x}_2 = x_2 \left( r_2 (1-g)^\alpha + d_2 g^\alpha \right) \left( \frac{x}{T} - 1 \right) \left( 1 - \frac{x}{K} \right) - c_2 x_2 - m_2 x_2 + m_1 x_1. \qquad (2)$$

For tumors exhibiting coordination games, the cancer cells' proliferation rates will be higher if the cellular composition is homogeneous. Furthermore, as the tumor grows its composition will converge on one or the other cell types as the predominant type. Both of these effects become stronger with a larger value for the convexity parameter. As the convexity parameter becomes larger so too does the punishment for discoordination. The mutation rate has the opposite effect. Increasing the mutation rate diversifies the composition of cancer cell types, reduces the fitness benefits achievable from coordination, and reduces the fitness of all of the cancer cells.

With respect to the composition of cancer, there exist two peaks of tumor fitness, at purely type 1 and at purely type 2. The extinction thresholds are lower and the maximum tumor sizes are higher when the tumor is homogeneous for one cell type or the other than when the tumor is heterogeneous (Fig 1).

Provided that the mutation rates are sufficiently small, a tumor that does not go extinct will achieve coordination on one or the other cancer cell type. In a typical calibration of parameters, the the tumor outside the extinction zone may converge to one of two stable equilibria, with either type 1 or type 2 becoming the predominant phenotype. There also exist four unstable equilibria: one for each phenotype on the border of the extinction zone and two mixed equilibria, one on the border of the extinction zone, and one with a high tumor burden (Fig 2).

## 3 Results

Coordination games of cancer can be leveraged to improve the outcomes of existing cancer therapies. In this section we model and simulate two types of therapy, one that increases the cell types' death rates, and one that increases their mutation rates.

In each case, as expected, one of the strategies has become dominant in the patient's tumor. Furthermore, the alternative strategy occurs at some very low frequency within the tumor as a result of recurrent mutations, epigenetic changes or other mechanisms for generating heritable phenotypic variation [35]. This key assumption has empirical merit given how mutator phenotypes are a hallmark of cancer, and the occurrence of large amounts of genetic variability has been shown to exist in virtually all tumors [36–38]. Furthermore, selection on cancer cells lines in vitro results in rapid evolution of novel phenotypes [39, 40].

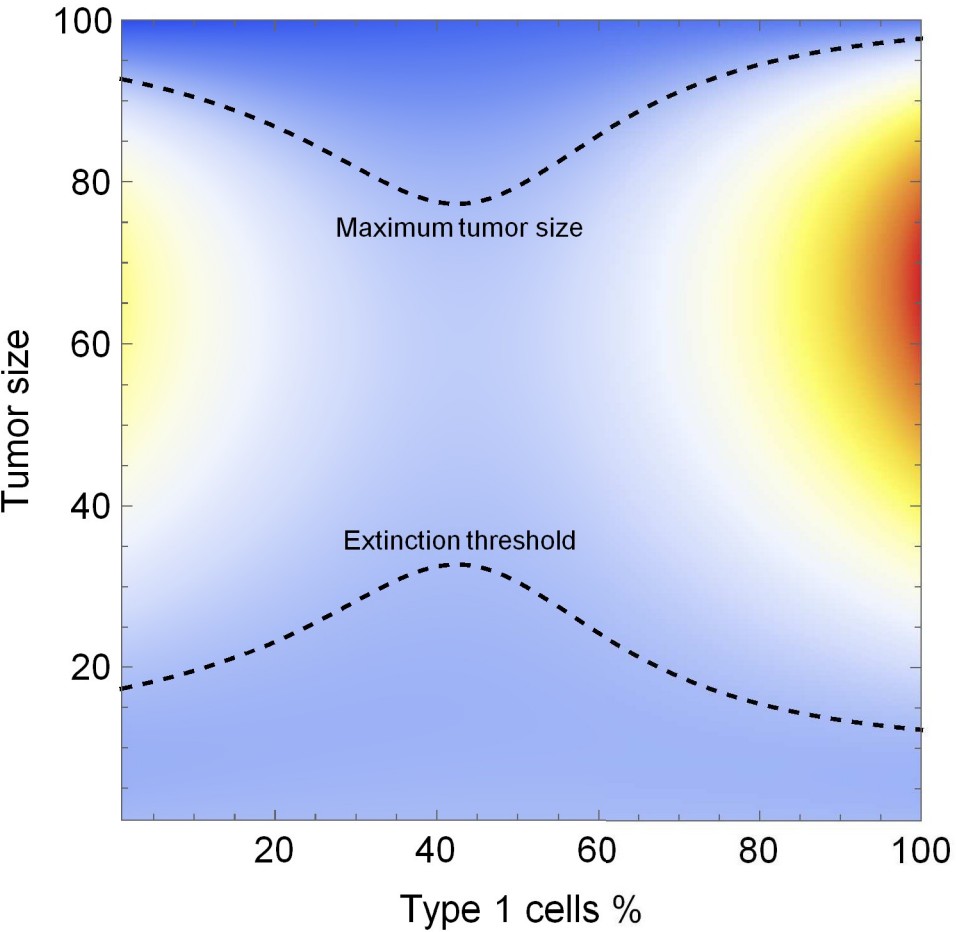

**Fig 1. The heat map of tumor growth.** Tumor growth, $\dot{x} = \dot{x}_1 + \dot{x}_2$ for the possible size-composition $(x, g)$ pairs, where the types' growth rates are modulated by a pure coordination game. Type 1's intrinsic growth rate, $r_1$, is larger and its death rate, $c_1$, is lower, therefore its fitness peak is higher, but a pure type 2 tumor also has a fitness peak. Parameters: $r_1 = 0.25$, $r_2 = 0.17$, $d_1 = d_2 = 0$, $K = 100$, $T = 10$, $\alpha = 2$, $c_1 = 0.05$, $c_2 = 0.1$, $m_1 = m_2 = 0.01$.

### 3.1 Coordination games and cytotoxic therapies

The challenge with many forms of cancer therapy such as chemotherapy is the evolution of resistance. Introducing a cytotoxic agent to attack the tumor yields a good initial response, with each subsequent use of the drug producing diminishing returns until the onset of resistance at which point the drug is ineffective.

In this section we model the onset of resistance as two strategies of a coordination game. Type 1 is *sensitive* to therapy, type 2 is *resistant*. We model the type's response to therapy by functions $(\gamma_1(t), \gamma_2(t))$ which add to the cells' death rates when the patient is receiving therapy. Compared to the base model of Section 1, we also omit the Allee-effect to better showcase the effect of leveraging the coordination game in eliminating the tumor. The model equations are then

$$\dot{x}_1 = x_1(r_1 g^\alpha + d_1(1-g)^\alpha)\left(1 - \frac{x}{K}\right) - (c_1 + \gamma_1(t))x_1 - m_1 x_1 + m_2 x_2, \tag{3}$$

$$\dot{x}_2 = x_2(r_2(1-g)^\alpha + d_2 g^\alpha)\left(1 - \frac{x}{K}\right) - (c_2 + \gamma_2(t))x_2 - m_2 x_2 + m_1 x_1. \tag{4}$$

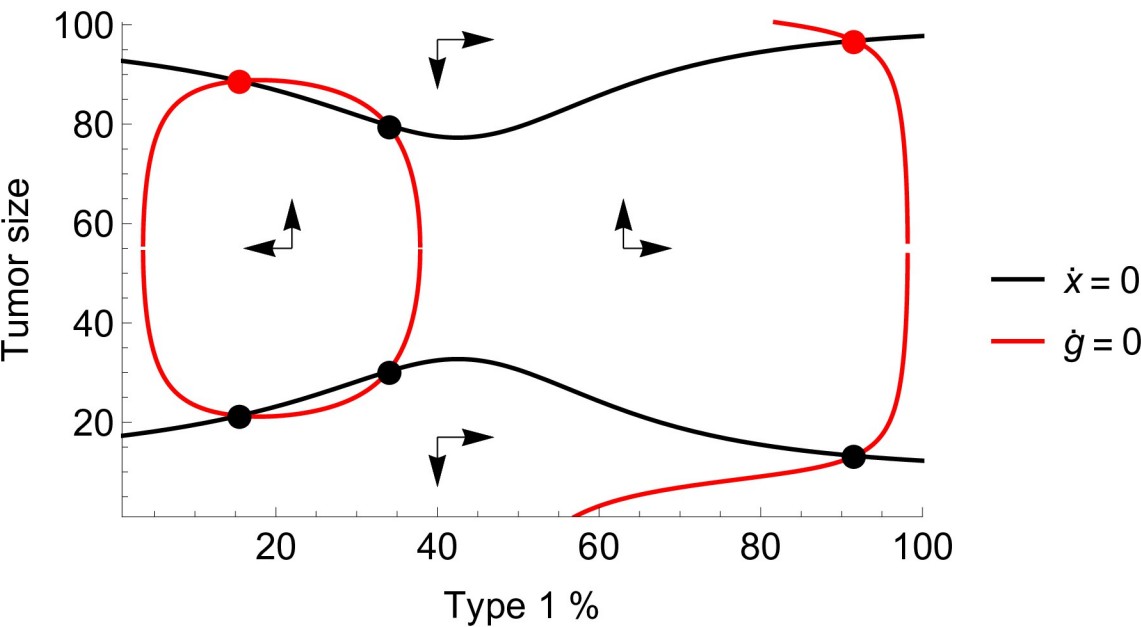

**Fig 2. Phase diagram of growth and composition.** Phase diagram of tumor growth and composition dynamics under a pure coordination game. There are two stable equilibria (red dots) and two corresponding basins of attraction for tumors above the extinction zone: either coordination is achieved on type 1, or type 2. Parameters: $r_1 = 0.25$, $r_2 = 0.17$, $d_1 = d_2 = 0$, $K = 100$, $T = 10$, $\alpha = 2$, $c_1 = 0.05$, $c_2 = 0.1$, $m_1 = m_2 = 0.01$.

For $i = 1, 2$ and we have $\gamma_i(t) = \gamma_i$ for time intervals when therapy is administered, indicating a constant dosage, with $\gamma_1 > \gamma_2 \geq 0$ and $\gamma_i(t) = 0$ when therapy is not administered. In the most extreme case, we have $\gamma_2 = 0$ indicating that type 2 is completely unaffected by the cytotoxic therapy. We illustrate the system in Fig 3.

Resistance sets in when an initially sensitive (type 1) tumor is disrupted by the therapy and achieves coordination on the resistant strategy (type 2). Upon initiation of therapy, the tumor size is reduced as type 1 cells are killed off. The tumor's composition becomes more and more mixed, further lowering its fitness and reinforcing the therapy. If therapy continues, the resistant type 2 cells will eventually come to predominate in the tumor and, in a competitive release from type 1 cells, the tumor population returns to high levels (left panel of Fig 4). Therapy may be turned off in hopes of avoiding the basin of attraction of the resistant type.

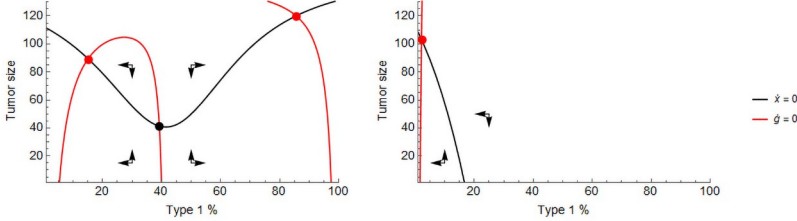

**Fig 3. Zero-isoclines with and without therapy.** The zero-isoclines of the system defined by (3) and (4) with therapy off (left) and on (right). If therapy is off, there are two stable equilibria (red dots) comprising mostly type 1 and mostly type 2 cells, respectively, and a mixed unstable equilibrium (black dot). With therapy on, the only stable equilibrium consists of mostly type 2 cells. Parameters: $r_1 = 0.4$, $r_2 = 0.2$, $d_1 = d_2 = 0$, $K = 150$, $\alpha = 2$, $c_1 = c_2 = 0.05$, $m_1 = m_2 = 0.01$, $\gamma_1 = 0.4$, $\gamma_2 = 0$.

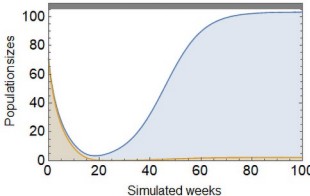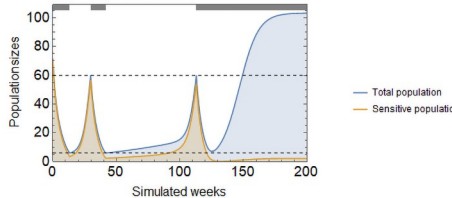

**Fig 4. Continuous and adaptive therapies.** Continuous cytotoxic therapy yields a good initial response, drives down tumor burden, but the population rebounds once cancer coordinates on the resistant type (left). Under adaptive therapy control over the tumor is lost when it enters the resistant basin of attraction, which happens when composition falls below $g$ = 40%, in the present case is after 3 treatment cycles (right). Parameters: $r_1 = 0.4$, $r_2 = 0.2$, $d_1 = d_2 = 0$, $K = 150$, $c_1 = c_2 = 0.05$, $m_1 = m_2 = 0.01$, $\alpha = 2$, $\gamma_1 = 0.4$, $\gamma_2 = 0$.

Adaptive therapies of cancer seek to retain control over the tumor by stopping therapy before resistance is complete, and restarting therapy only when the tumor burden becomes threatening. In a coordination game, adaptive therapy must strike a balance between staying within the sensitive type's basin of attraction to maintain control, and avoiding dangerous levels of the tumor burden. In the right panel of Fig 4, the cytotoxic therapy is turned off when the tumor burden falls below 6 and turned on again when it exceeds 60. Control over the tumor is maintained for three on-off cycles.

Making use of the coordination game can improve therapy outcomes. As seen in Fig 4, mixed tumors have a slow rate of growth even when the tumor is small. Therapy regimens can be designed to keep the tumor composition mixed as follows: have cytotoxic therapy on while it has an effect but does not leave the sensitive type's basin of attraction. Before transiting to the resistant basin of attraction, therapy is turned off. The tumor size begins growing again, and before it exceeds its starting point, therapy is turned on again. If tumor composition can be monitored perfectly, then such a therapy can lead to the extinction of the tumor (Fig 5 left panel).

Practically, conditioning treatment on the tumor's composition is highly problematic as the prevalence of resistance can only be estimated from data on the tumor's response to the cytotoxic therapy. In case of adaptive therapy in practice, decisions on stopping or restarting therapy are based on tumor size. Treatment regimens conditioned on tumor size are, predictably, much less reliable in controlling both the tumor size and its composition and therefore produce more modest results. Yet, a broad range of therapy strategies exist that keep the tumor in a recurrent, persistent cycle (Fig 5, right panel). In SI.4 we discuss the advantages and disadvantages of four different strategies, each being a trade-off between practicality of execution and safety from losing control over the tumor.

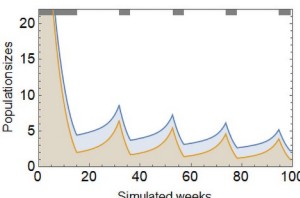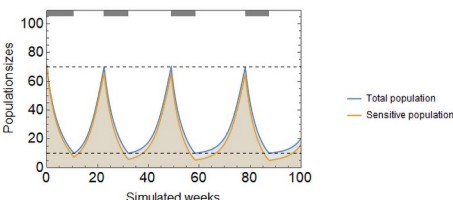

**Fig 5. Adaptive therapy conditioned on tumor size and composition.** Cytotoxic therapy conditioned on the tumor's composition can lead to the extinction of the tumor (left), while therapy conditioned on tumor size can lead to persistent cycles (right). Therapy is turned off when $g(t)$ ($x(t)$) falls below the lower bound $\underline{g}$ ($\underline{x}$) and it is turned on again when it rises above the upper bound $\bar{g}$ ($\bar{x}$). Parameters: $r_1 = 0.4$, $r_2 = 0.2$, $d_1 = d_2 = 0$, $K = 150$, $c_1 = c_2 = 0.05$, $m_1 = m_2 = 0.01$, $\alpha = 2$, $\gamma_1 = 0.4$, $\gamma_2 = 0$, $\underline{g} = 45\%$, $\bar{g} = 75\%$, $\underline{x} = 10$, $\bar{x} = 70$.

## 3.2 Mutation-promoting therapy

As shown by the previous section, coordination games may be leveraged for improving cytotoxic therapy outcomes. In the extreme, adaptive therapy in the context of a coordination game can engineer an extinction following a certain number of on-off cycles of the drug. More practically, on-off adaptive therapy strategies can indefinitely maintain a non-lethal tumor burden with a controllable composition. Up until this point, the only tool in the simulated physician's arsenal was cytotoxic therapy, i.e. increasing the sensitive cell type's death rate.

A more straightforward method for taking advantage of the coordination game involves lowering tumor fitness by maintaining a mixed composition of cancer cell types. One way to do this is by promoting mutation between the two types. We model this by adding type-specific response functions ($\sigma_1(t)$, $\sigma_2(t)$) which add to the cells' mutation rates while the patient is under this type of therapy, called mutation therapy. Together with cytotoxic therapy, the model equations become

$$\dot{x}_1 = x_1\big(r_1 g^\alpha + d_1(1-g)^\alpha\big)\Big(1 - \frac{x}{K}\Big) - (c_1 + \gamma_1(t))x_1 - (m_1 + \mu_1(t))x_1 + (m_2 + \mu_2(t))x_2, \quad (5)$$

$$\dot{x}_2 = x_2\big(r_2(1-g)^\alpha + d_2 g^\alpha\big)\Big(1 - \frac{x}{K}\Big) - (c_2 + \gamma_2(t))x_2 - (m_2 + \mu_2(t))x_2 + (m_1 + \mu_1(t))x_1. \quad (6)$$

For $i = 1, 2$ and we have $\mu_i(t) = \mu_i$ for time intervals under mutation therapy, with $\mu_1, \mu_2 \geq 0$, and $\mu_i(t) = 0$ otherwise. Crucially, we allow both types' mutation therapies, and cytotoxic therapy to be administered at different times, i.e. at any time the patient may receive any combination of the three, or may receive no therapy at all.

The role of mutation therapy is not to attack the tumor directly, but rather, to achieve or maintain a heterogeneous composition of cell types so that the tumor's growth is inhibited by the mechanics of the coordination game, and direct attacks launched by other means are more effective. Mutation therapy pushes the stable composition isocline curve inwards, thus the stable equilibria will be more heterogeneous. If mutation therapy is strong enough, it may eliminate one of the stable equilibria, forcing the tumor to coordinate on a selected type. In Fig 6 we showcase the effects of mutation therapy in the system defined by (5) and (6).

In the sensitive-resistant model, this effect can be exploited to force coordination on the sensitive type by mutating the resistant type to a sensitive one. We identify three ways by which doing so would improve therapy outcomes. First, applying continuous cytotoxic therapy together with mutation therapy delays coordination on the resistant type resulting in a longer progression time than without mutation therapy. Second, applying cytotoxic therapy adaptively is made safer with mutation therapy as the resistant type's basin of attraction becomes smaller, thus the risk of losing control is greatly reduced. Third, new types of therapy aimed at the extinction of the tumor become possible. Fig 7 shows each of these in turn: (1) mutation therapy delays tumor progression under continuous cytotoxic therapy (top left panel), (2) mutation therapy makes it possible for a successful adaptive therapy to reach as low as $g = 32\%$ without losing control over the tumor (top right), and (3) mutation therapy makes it possible to engineer a successful extinction therapy (bottom right) that otherwise would be impossible in its absence (bottom left).

## 4 Discussion

In this paper we advanced the game theory literature of cancer by proposing that coordination games, a basic class of games in non-cooperative game theory, may exist in evolutionary games of cancer. Populations playing a coordination games tend to converge on a single predominant

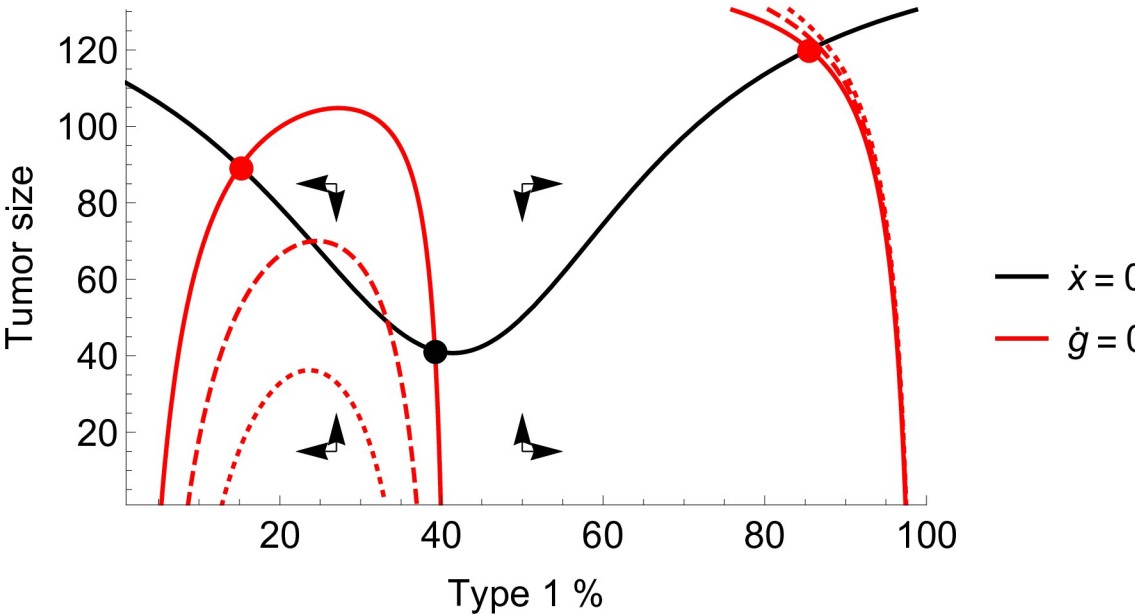

**Fig 6. Zero-isoclines under mutation therapy.** The effects of mutation therapy on the system's phase diagram. Mutating type 2 cells to type 1 cells shrinks the former's basin of attraction. A slight mutation therapy maintains two stable equilibria, one for each type (dashed red), a stronger one eliminates the type 2-dominated stable equilibrium (dotted red). Parameters: $r_1 = 0.4$, $r_2 = 0.2$, $d_1 = d_2 = 0$, $K = 150$, $c_1 = c_2 = 0.05$, $m_1 = m_2 = 0.01$, $\alpha = 2$, solid red: $\mu_1 = 0$, dashed red: $\mu_2 = 0.005$, dotted red: $\mu_2 = 0.01$.

phenotype, possibly eliminating any competing phenotypes, hence observing such an evolutionary game in its later stages obscures any alternate strategies. As a result, evolutionary coordination games may go unnoticed. We proposed a number of examples that hint at the existence of coordination games in cancer and identified the conditions under which they appear in cancer.

In capturing coordination games in cancer, our paper introduces two technical novelties to Lotka-Volterra competition models. First, we replace the exogenous growth parameter 'R'

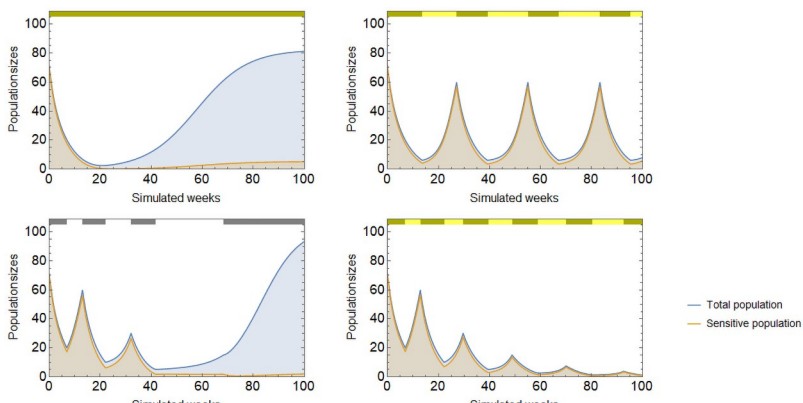

**Fig 7. Mutation therapy together with cytotoxic therapy.** Mutation therapy when applied with cytotoxic therapy improves therapy outcomes. Continuous cytotoxic therapy combined with mutation therapy delays coordination on the resistant type (top left). Adaptive cytotoxic therapy combined with mutation therapy may reach lower composition ($g = 32\%$) without losing control (top right). Therapy aimed at the extinction of the tumor fails without mutation therapy (bottom left) but succeeds with it (bottom right). Parameters: $r_1 = 0.4$, $r_2 = 0.2$, $d_1 = d_2 = 0$, $K = 150$, $c_1 = c_2 = 0.05$, $m_1 = m_2 = 0.01$, $\alpha = 2$, $\gamma_1 = 0.4$, $\gamma_2 = 0$, $\mu_1 = 0.02$, $\mu_2 = 0$.

with endogenous growth derived from an evolutionary coordination game. Second, we scale the punishment of discoordination through the convexity parameter (a biological interpretation of which is provided in SI. 3). As a result, most cancer growth occurs when the tumor is coordinated on either of the two strategies ([Fig 1]).

Coordination games in cancer may be exploited in therapy to the benefit of the patient. If the competing phenotypes react differently to therapies, the treating physician has the option to affect the balance between them. Maintaining a heterogeneous composition of two competing phenotypes within the tumor reduces its fitness. In idealized conditions this may make the difference between a successful therapy that ends with the tumor's extinction and a failed one. In more realistic conditions adaptive therapy regimens may be designed to take advantage of coordination game by keeping the tumor's size and composition under control, possibly indefinitely. Finally, if two competing phenotypes may be mutated into one another, new therapy regimens may be designed through which a heterogeneous composition of the tumor is maintained directly by the treating physician.

We raise three remarks, two technical and one conceptual. First, in this paper we primarily considered coordination games with two competing phenotypes. This is done primarily for expository and graphical purposes and can be generalized to include any number of phenotypes. Multi-strategy coordination games are characterized by payoff matrices where each entry of the principal diagonal is larger than any other entry in the same row or column. That is, each strategy's fitness is maximized in its own environment, and each strategy accommodates itself more than it does any other strategy. Each such game, however, reduced to any two of its strategies produces a two-strategy coordination game, thus identifying two-strategy coordination games is necessary for multi-strategy ones.

Second, in our simulations we assumed a sensitive phenotype and a resistant one. This is not to suggest that phenotypes of coordination games in cancer necessarily have such a stark contrast in their response to cytotoxic therapy. Nor do we wish to make a pretense that coordination games are behind resistance mechanisms to cancer therapy. Instead, this is to showcase that, given a different response to a cytotoxic agent by two strategies of a coordination game, the physician may take advantage of the underlying game. Any difference between the response rates may be exploited to try to maintain a heterogeneous composition of the tumor.

Finally, we mention cancer heterogeneity. It is well known that tumors display a high degree of genotypic and phenotypic heterogeneity and a rapid rate of mutation that increases heterogeneity rather than decreases it as the disease progresses. This seems in contradiction with our hypothesis that coordination games exist in cancer since those would induce a move towards homogeneity in the tumor. We argue, however, that this feature of cancer does not rule out the possibility of coordination games. The strategies of evolutionary games in cancer are generally not meant to represent a single phenotype, but usually a large number of them that coalesce into an observable trait of the tumor. We propose that coordination games take place between these traits. To continue the "rules of the road" analogy, just because the cars all drive on the right or the left side, the cars themselves may still be diverse, and even continue to branch out to display more and more heterogeneity.

## Supporting information

**S1 File.**
(PDF)

## Acknowledgments

We thank Ingela Alger, David Basanta, Jorge Peña, Monica Salvioli, and Jeffrey West, as well as seminar participants of Moffitt's Integrated Mathematical Oncology Department and the Institute of Advance Studies in Toulouse for useful discussions and comments. The authors declare no potential conflicts of interest.

## Author Contributions

**Conceptualization:** Péter Bayer, Joel S. Brown.

**Formal analysis:** Péter Bayer.

**Investigation:** Péter Bayer, Patricia H. McDonald, Derek R. Duckett, Kateřina Staňková, Joel S. Brown.

**Methodology:** Péter Bayer, Kateřina Staňková.

**Supervision:** Joel S. Brown.

**Validation:** Robert A. Gatenby.

**Visualization:** Péter Bayer, Kateřina Staňková.

**Writing – original draft:** Péter Bayer.

**Writing – review & editing:** Péter Bayer, Robert A. Gatenby, Patricia H. McDonald, Derek R. Duckett, Kateřina Staňková, Joel S. Brown.

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
