## [Decision Letter · Decision Letter 0]

4 Oct 2021

PONE-D-21-26125Coordination games in cancerPLOS ONE

Dear Dr. Bayer,

Thank you for submitting your manuscript to PLOS ONE. After careful consideration, we feel that it has merit but does not fully meet PLOS ONE’s publication criteria as it currently stands. Therefore, we invite you to submit a revised version of the manuscript that addresses the points raised during the review process.

We look forward to receiving your revised manuscript.

Kind regards,

Jun Tanimoto

Academic Editor

PLOS ONE

Journal Requirements:

Reviewers' comments:

Reviewer's Responses to Questions

**Comments to the Author**

1. Is the manuscript technically sound, and do the data support the conclusions?

Reviewer #1: Yes

Reviewer #2: Partly

2. Has the statistical analysis been performed appropriately and rigorously? 

Reviewer #1: Yes

Reviewer #2: N/A

3. Have the authors made all data underlying the findings in their manuscript fully available?

Reviewer #1: Yes

Reviewer #2: Yes

4. Is the manuscript presented in an intelligible fashion and written in standard English?

Reviewer #1: Yes

Reviewer #2: Yes

5. Review Comments to the Author

Reviewer #1: Motivated by medical science, this work apply the concept of coordination game to a specific competing situation between cancer cell and treatment (patient) with tumor growth. The authors explored many examples along this concept, which makes me somehow interesting.

Yet, honestly, I must say that this work has less amount of novelty in terms of scientific finding.

I can agree, however, that this article might bring practically meaningful information to some audience. In line with this, I rather have a positive feeling on the MS.

One point, which I must suggest the authors when revising to a revised MS, is that Introduction part should contain some important recent findings of evolutionary game theory. That is; the universal concept of dilemma strength in 2 by 2 game including coordination game.

As the authors saying the current MS, what some previous studies presuming to adopt the game framework into tumor cell growth is Prisoner’s dilemma (PD) or Chicken game (other favor to call Snowdrift game). Needless to say, a coordination game belongs to Stag Hunt (SH) game. In general for a symmetric 2 by 2game, there are two dilemmas; Chicken-type dilemma (of which extent is quantified by Dg’ := (T – R)/(R – P)) and Stag Hunt-type dilemma (of which extent is quantified by Dg’ := (P – S)/(R – P)), PD is simultaneously suffered from both dilemmas (Dg’ >0 & Dr’>0). In contrast, SH game is feature only with Stag Hunt-type dilemma; Dg’<0 & Dr’>0.

I strongly suggest the authors to review the concept of dilemma strength in view of ‘coordination game’ in Introduction part by citing some relevant literature; (i) Dilemma strength as a framework for advancing evolutionary game theory: Reply to comments on “Universal scaling for the dilemma strength in evolutionary games”, Physics of Life Reviews 14, 56-58, 2015, (ii) Scaling the phase- planes of social dilemma strengths shows game-class changes in the five rules governing the evolution of cooperation, Royal Society Open Science, 181085, 2018, (iii) Social efficiency deficit deciphers social dilemmas, Scientific Reports 10, 16092, 2020.

Reviewer #2: Reviewer comments

This paper describes about game theory in cancer cells. They treated the elements of the payoff matrix in coordination game as a parameter of the Lotka-Volterra (LV) competition model. They propose a strategy to prevent the growth of tumors by encouraging the coexistence of two types of cancer cells through therapy.

This study sounds interesting. However, I cannot recommend acceptance of the paper in current states.

Comment 1.

I cannot understand the validity of the model.

In both the lung and breast cancer explanations, it is unclear whether game-like dynamics really exist in the background. In other words, I don't think these explanations provide examples of what cancer dynamics this model can be applied to. These sentences are argumentative and don't seem to motivate the development of the model.

The author also wrote as follows:

“The alternative strategies that the population could have coordinated upon are no longer visible. If coordination games exist in cancer, by the time the disease is detected, most cancer patients would likely present with a cancer that has already evolved to a common phenotype.”

Does above sentences mean that we don't know if there is a coordination game going on in cancer? In addition, it seems that at the time a cancer is discovered in the patient, a single type of cancer is already dominant and “Treatment with two types of competitive systems” as the authors assume is not possible. If so, what is the purpose of this model?

Comment 2.

In introduction section,

“As a result, the various strategies of these games are more likely to manifest between rather than within patients.”

I cannot catch up the meaning of this sentence.

Comment 3.

I am skeptical that the current title adequately describes this study, because the current model looks like a traditional LV model of 2-species competition system. Where is the game in that? The elements of payoff matrix are just used as part of the parameters of the LV model.

Comment 4.

Furthermore, I felt that the author ignored the vast amount of knowledge and previous studies on the LV competition equation. As far as I know that LV competition model represent “competitive exclusion principle”, where we can see the elimination of species and the survival of only one species without the assumption of coordination game concept. Therefore, from the perspective of ecology, which has been trying to understand coexistence mechanisms, this model seems to be a derivative of many previous multispecies coexistence models. So, I have not yet to discover the uniqueness and aim of this model. This comment is another reason why I think the title is not match the content.

Comment 5.

It is difficult to understand the meaning of figures. In Fig.1,

Comment 5-1

the author wrote “The heat map of tumor growth modulated by a pure coordination game”, what is “tumor growth”? Is this “x[t=T]/x[t=0]” when the final time T?

Comment 5-2

I cannot see two red dots in fig.1.

Comment 5-3.

Are the vertical and horizontal axes means initial values?

Comment 5-3.

What's going on with m1 and m2? (maybe m1=m2=0?)

Comment 5-4.

The red part on the right is the x1 mountain and the red part on the left is the x2 mountain, is it right? I didn't know this until I did the calculation myself. I'm not sure if it's okay to show two different types of cancer growth in the same system of colors.

I conclude the rejection for the above reasons.

6. PLOS authors have the option to publish the peer review history of their article (what does this mean?). If published, this will include your full peer review and any attached files.

Reviewer #1: No

Reviewer #2: No

---

## [Author Response · Author response to Decision Letter 0]

1 Dec 2021

We thank the reviewers for their efforts in commenting on our manuscript. Please see our 'Response to reviewers' attachment for detailed responses on your comments.

---

## [Decision Letter · Decision Letter 1]

6 Dec 2021

Coordination games in cancer

PONE-D-21-26125R1

Dear Dr. Bayer,

We’re pleased to inform you that your manuscript has been judged scientifically suitable for publication and will be formally accepted for publication once it meets all outstanding technical requirements.

Kind regards,

Jun Tanimoto

Academic Editor

PLOS ONE

Additional Editor Comments (optional):

Reviewers' comments:

Reviewer's Responses to Questions

**Comments to the Author**

1. If the authors have adequately addressed your comments raised in a previous round of review and you feel that this manuscript is now acceptable for publication, you may indicate that here to bypass the “Comments to the Author” section, enter your conflict of interest statement in the “Confidential to Editor” section, and submit your "Accept" recommendation.

Reviewer #1: All comments have been addressed

2. Is the manuscript technically sound, and do the data support the conclusions?

Reviewer #1: Yes

3. Has the statistical analysis been performed appropriately and rigorously? 

Reviewer #1: Yes

4. Have the authors made all data underlying the findings in their manuscript fully available?

Reviewer #1: Yes

5. Is the manuscript presented in an intelligible fashion and written in standard English?

Reviewer #1: Yes

6. Review Comments to the Author

Reviewer #1: Although another reviewer did suggest Rejection, the part I suggested was fairly responded by the revised MS. Thus, now I could agree with publication.

7. PLOS authors have the option to publish the peer review history of their article (what does this mean?). If published, this will include your full peer review and any attached files.

Reviewer #1: No

---

## [Editor Report · Acceptance letter]

28 Dec 2021

PONE-D-21-26125R1 

Coordination games in cancer 

Dear Dr. Bayer:

I'm pleased to inform you that your manuscript has been deemed suitable for publication in PLOS ONE. Congratulations! Your manuscript is now with our production department. 

Kind regards, 

on behalf of

Prof. Jun Tanimoto 

Academic Editor

PLOS ONE